# Application of Single-Frequency Arbitrarily Directed Split Beam Metasurface Reflector in Refractive Index Measurements

**DOI:** 10.3390/s24206519

**Published:** 2024-10-10

**Authors:** Brian M. Wells, Joseph F. Tripp, Nicholas W. Krupa, Andrew J. Rittenberg, Richard J. Williams

**Affiliations:** 1Department of Physics, University of Hartford, West Hartford, CT 06117, USA; rittenber@hartford.edu (A.J.R.); ricwillia@hartford.edu (R.J.W.); 2CT State Community College, Norwalk, CT 06854, USA; jtrip0005@mail.ct.edu; 3Department of Electrical Engineering, University of Hartford, West Hartford, CT 06117, USA; nkrupa@hartford.edu

**Keywords:** metasurface, metamaterials, microwave optics, transformation optics, index of refraction, 3D printing

## Abstract

We present a sensor that utilizes a modified single-frequency split beam metasurface reflector to measure the refractive index of materials ranging from one to three. Samples are placed into a cavity between a PCB-etched dielectric and a reflecting ground plane. It is illuminated using a 10.525 GHz free-space transmit horn with reflecting angles measured by sweeping a receiving horn around the setup. Predetermined changes in measured angles determined through simulations will coincide with the material’s index. The sensor is designed using the Fourier transform method of array synthesis and verified with FEM simulations. The device is fabricated using PCB milling and 3D printing. The quality of the sensor is verified by characterizing 3D printed dielectric samples of various infill percentages and thicknesses. Without changing the metasurface design, the sensing performance is extended to accommodate larger sample thicknesses by including a modified 3D printed fish-eye lens mounted in front of the beam splitter; this helps to exaggerate changes in reflected angles for those samples. All the methods presented are in agreement and verified with single-frequency index measurements using Snell’s law. This device may offer a viable alternative to traditional index characterization methods, which often require large sample sizes for single-frequency measurements or expensive equipment for multi-frequency parameter extraction.

## 1. Introduction

Metasurfaces are two-dimensional (2D) arrangements of ultra-thin, sub-wavelength scatterers, called “meta-atoms”, that can be engineered to elicit exotic electromagnetic properties. Metasurfaces may take the form of periodic or aperiodic arrays, with applications in antennas [1,2,3], vortex generation [4,5], electromagnetic (EM) absorption [6,7], flat lenses [8,9], and sensing [10]. Recently, metasurfaces have been employed as refractive index sensors by changing the absorption response of a metasurface in the presence of an unknown sample [11], altering the polarization conversion efficiency of the metasurface due to an unknown liquid [12], and spectral resonance shifts [13].

In this work, we present an alternative mechanism for refractive index sensing of an unknown sample based on beam deflection from a metasurface beam splitter. Many of the established methods available to measure the refractive index require sweeping a frequency source to observe shifts in resonant frequencies [14,15,16] or to obtain free-space measurements required for parameter extraction [17]. Both of these approaches require costly equipment such as a vector network analyzer (VNA). The approach presented in this work operates at a single frequency utilizing two scattered beams from a metasurface reflective beam splitter; thus, the measured angle deviations of the reflected split beams of an unknown sample can be correlated to the sample’s refractive index. A low-cost, single-frequency, beam splitter-based metasurface refractive index sensor was fabricated via PCB milling and 3D printing. The sensor’s refractive index sensing capabilities were verified through finite element (FEM) simulations and experimentally with a free-space measurement setup for samples with refractive indices ranging from one to three.

The paper is structured into the following five main sections: (1) an overview of the theoretical principles underpinning the Fourier transform method of array synthesis as it relates to the design of arbitrarily directed split-beam metasurface reflectors, (2) a detailed explanation of the design process and verification of the metasurface sensor, incorporating both theoretical analysis and finite element method (FEM) simulations, (3) comprehensive insights into the fabrication process and the experimental setup employed in the study, (4) presentation of the measured results and experimental verification procedures, and (5) an in-depth discussion of the findings and their implications.

## 2. Theory

The metasurface refractive index sensor is designed using a technique developed from the Fourier transform method of array synthesis [18]. The inverse Fourier transform of a desired far-field scattering pattern is performed to yield the complex reflection coefficients of the metasurface beam-splitting reflector. This technique has been previously employed to successfully design a metasurface beam splitter that reflected a normally incident wave to two arbitrarily chosen scattering angles [19].

The array factor gives the desired far-field scattering pattern of the metasurface, and for the case of a metasurface beam splitter, the far-field pattern is described by the sum of the two-dimensional delta functions in (u,v) space.
(1)$$F\left(u,v\right)=\frac{1}{N_{beams}}\sum_{i=1}^{N_{beams}}\delta (u-u_{i})\delta (v-v_{i})$$The delta functions represent the scattered beams, with $\left(u_{i},v_{i}\right)$ representing the scattering direction of beam i. The inverse Fourier transform of Equation (1) yields the complex reflection coefficients of each meta-atom composing the metasurface beam splitter.
(2)$$a_{mn}=\frac{d_{x}d_{y}}{\lambda^{2}}\int_{-\frac{\lambda }{2d_{x}}}^{\frac{\lambda }{2d_{x}}}\int_{-\frac{\lambda }{2d_{y}}}^{\frac{\lambda }{2d_{y}}}F\left(u,v\right)e^{-ik(md_{x}u+nd_{y}v)}dudv$$The real component of Equation (2) describes the magnitude of the reflection coefficients distributed across the metasurface, while the complex argument of Equation (2) describes the reflection phase distribution across the metasurface.

## 3. Design

The realization of the metasurface beam splitter sensor begins with finite element method (FEM) unit cell simulations using COMSOL Multiphysics for various conductive patch configurations. This step is necessary to create a selection library of patch geometries for our theoretical model. The simulations involve changing the size of conductive rectangular patches on a dielectric substrate above a sample cavity and reflector (Figure 1a–c). We use a normally incident 10.525 GHz, x-polarized plane wave to illuminate the unit cell with periodic boundaries. The unit cell is a square with a periodicity of p=9.3 mm, and it is made up of a 30 μm thick conductive patch of $1.25\times 10^{6}\;S/m$ atop a 0.8 mm thick dielectric with a relative permittivity of 4.8. Below the substrate, there is a 1 mm thick air gap above a 200 μm conductive plate of $3.77\times 10^{7}\;S/m$. The $l_{x}$ and $l_{y}$ side lengths of the conductive patch are varied from 50 μm  to 9.3 mm, and reflection coefficients are calculated from FEM simulations. Figure 1b,c show the layout of the unit cell, and Figure 1d,e depict the reflection properties of the metasurface unit cell with varying patch dimensions. The period of the unit cell is selected to ensure that the different sizes of the conductive rectangular patches are within the fabrication tolerances of our PCB milling machine while also providing the necessary reflection properties. The values for conductivity and permittivity are chosen or were measured using a four-point probe to best match the materials used for fabricating the final design.

For the metasurface beam splitter reflector design, we chose to use an 8 × 8 conductive patch array to cast two symmetric reflected waves at angles ($\theta_{R}^{\prime }=-30^{{}^{\circ}}$, $\phi_{R}^{\prime }=0$) and ($\theta_{R}=30^{{}^{\circ}}$, $\phi_{R}=0$) via a normally horizontal polarized incident beam. This particular geometry and design are selected for the following reasons: (1) to have the most diminutive effective dimensions to reduce the overall sample size, (2) to simplify the experimental measuring process by having both reflected beams be on the same plane, and (3) to facilitate the detection of any reflected angle shifts due to changes in the sample’s material indices by implementing symmetric beam scattering. The shifts for both reflected beams will also be symmetric.

Additionally, the size and functionality of the fabricated metasurface refractive index sensor are influenced by the beam pointing error due to amplitude and phase errors across the metasurface [20]. In this scheme, amplitude and phase errors manifest from the presence of the unknown dielectric samples and their imparted amplitude and phase mismatches relative to the sample-free sensor. The impact of the beam pointing error relative to the beam pointing error of a sample-free sensor as a function of sensor size is shown in Figure 2e for dielectric samples with indices of refraction ranging from 1.5 to 3. As shown, for the 8 × 8 metasurface sensor, the sample under test (SUT) has a larger impact on the beam pointing error, translating into more considerable beam deflections and improving the ability to measure differences in refractive indices of SUTs.

The patch design of the metasurface is determined by the complex reflection coefficients distributed across the metasurface using Equations (1) and (2). A nearest-neighbor search is performed using the simulated unit cell data to find the best $l_{x}$ and $l_{y}$ patch lengths that will produce the most similar results for the desired beam scattering. This is performed using a Euclidean distance metric on a two-dimensional space, where the reflection magnitude is on one axis and the reflection phase is on the other. The distance calculated is the distance between the desired reflection and the simulated reflection properties. For this particular case, the surface reflection and phase magnitude will be uniform throughout the y-direction, and only the x-direction needs to be calculated. This is a consequence of the reflected beams being scattered in the same plane; these results are illustrated in Figure 2a,b.

After the patch geometries are determined (Figure 2c), FEM simulations are conducted using COMSOL Multiphysics to verify that the design accurately produces the desired beam-splitting qualities. These results are presented in Figure 2d and show good agreement between all three approaches, namely the theoretical approach, patch design, and FEM simulations. The sensing characteristics of the beam splitter are studied by systematically changing the refraction index of the 1 mm sample from 1 to 3 in 0.05 increments and measuring the displacement of the reflected angles’ peak maximums from its initial position (Figure 3). A maximum displacement of 8° is observed with an approximately 0.5° variation for every 0.05 index change. Using these results to compare the change in the peak of the reflected angles from both beams will allow for the determination of an unknown material’s index between one and three.

## 4. Fabrication and Experimental Setup

The metasurface beam splitter was fabricated by etching the array pattern on a 0.8 mm MG Chemicals FR4 single-sided 1 oz copper-clad laminate circuit board using a modified Sainsmart Genmitsu 3020-PRO MAX V2 CNC Router Machine with a 0.8 mm flat-end corncob titanium coat mill bit. The final fabricated model is illustrated in Figure 4a. A custom sensor holder was 3D printed using Polymaker PolyLite PLA to hold the aluminum ground plane reflector, samples, and etched metasurface (Figure 4b). The design of the sensor holder allows for the easy removal and swapping of samples during experimental measurements. The samples were 3D printed using natural Polymaker PolyLite PLA, with gyroidal infills ranging from 10% to 100% (Figure 4c) at four different sample thicknesses from 1 mm to 4 mm. In total, 44 samples were experimentally measured, and their indices were determined.

The optical properties of the samples were measured using horn antennas from the PASCO microwave optics system (WA-9314C). The Gunn diode transmitter (WA-9801) was secured to a boom stand and positioned for maximum sensor illumination; this was determined to be $3\lambda_{0}$ from the front of the beam splitter. The receiving antenna (WA-9800) was mounted to a motorized linear guide arm affixed at a position of 60 cm from the sensor. The linear guide arm can be rotated at 0.01° increments using a motorized rotating optical platform centrally positioned beneath a stationary sample platform (Figure 4d). The radial and rotational positions were controlled using an Arduino IDE and Longruner GRBL CNC shield board interfaced with MATLAB. The transmitter used a low-voltage source to produce linearly polarized microwaves with a fixed frequency of 10.525 GHz at 15 mW. The receiver had a built-in amplifier with a sensitivity scale, which was externally connected to a 4½ digit B&K Precision 2831E tabletop digital multimeter controlled using MATLAB R2024a. The receiver was mechanically moved at 0.1° increments ±20° from the measured maximum reflected peaks. The relative electric field was measured at each theta position by averaging 3 s or 75 points of multimeter voltage data. Pyramidal foam absorbers were positioned about the measurement area to reduce stray scattering from the surroundings.

The maximum reflecting angles of each sample were measured, and the difference in the reflected angle was calculated. The change in the reflected angle was calculated by subtracting the reflected angle measured with a sample from the no-sample configuration, which included only an air gap between the substrate and reflector. The change in reflected angles created by the varying samples was compared to the FEM simulated results of the same design. The experimental results could then be fit to the FEM simulated values for each index, and the sample’s measured index could be extracted.

To verify the measurements obtained from the metasurface beam-splitting reflecting sensor, experimental and theoretical calculations of the index of refraction as a function of infill percentage were determined using Snell’s Law. A 25° prism was designed and fabricated using natural Polymaker PolyLite PLA for the following six infill percentages: 10%, 20%, 30%, 50%, 75%, and 100%. By measuring the signal strength from the prisms at given angular positions along the receiver’s path, as shown in Figure 5, the index of refraction $n_{p}$ is calculated using
(3)$$\;\;n_{p}\sin\left(\theta_{p}\right)=n_{air}\sin\left(\theta_{p}+\theta_{m}\right)$$
where the angle of the prism is $\theta_{p}$ and the measured angle from the RX is $\theta_{m}$. The same prism geometry with infills was also simulated using COMSOL Multiphysics. An equivalent methodology was used to calculate the index of refraction from the electric field profiles produced via the simulation. The experimental and theoretical results are plotted in Figure 5c and exhibit excellent agreement. A polynomial fit was calculated from these results, producing a function that relates the infill percentage and the index of refraction.
(4)$$n\left(p\right)=-3.683\times 10^{-6}p^{2}+0.0064p+0.9908$$The above equation allows for the calculation of the index of refraction at any infill percentage for the natural Polymaker PolyLite PLA, where n refers to the index of refraction and p  is the infill percentage. This result is used to verify the measurements obtained from the metasurface beam splitter reflecting index sensor.

## 5. Results

To verify the effectiveness of the metasurface beamsplitter sensor, experimental measurements and index calculations are performed on 44 different samples fabricated using natural Polymaker PolyLite PLA. These samples had varying gyroidal infill percentages, ranging from 0% to 100% in increments of 10%, with four thickness types as follows: 1 mm, 2 mm, 3 mm, and 4 mm, equivalent to $0.035\lambda_{0}$, $0.070\lambda_{0}$, $0.105\lambda_{0}$, and $0.140\lambda_{0}$. The first thickness to be investigated is 1 mm. These samples have the thinnest profile and provide a 98.7% reduction in mass when compared to the prisms. As outlined in the experimental setup, the electric far fields for this thickness are measured using 0.1° angular increments ±20° from the measured maximum reflected peaks with the receiving horn fixed at a radial distance of 60 cm from the metasurface. The maximum reflected angular position of each infill is determined by locating the maximum value from the electric-far-field profiles for both the −30° and +30° reflected angles, which is illustrated in Figure 6a. A polynomial fit is performed on the FEM simulations for the change in angle between the sensor without a sample to all measured peak locations with the samples present, Δθ, and the sample’s index of refraction (Figure 6b). The changes in angles are used rather than the maximum reflected angles to allow for potential variation in comparing measured maximum angles between the simulation and experiment. Δθ is calculated for the 1 mm measured samples, and the index for each infill is determined using the polynomial fit obtained from the FEM simulations. These results are compared to the FEM simulations (Figure 6c) and the prism measurements (Figure 6d).

The index of refractions obtained from the metasurface beam-splitting sensor for the 1 mm samples are in good agreement with the prism measurements; at its highest, it has a 6.7% error for the 30% infill, and all other infills are within no more than a 3% error.

The next samples to be characterized are the 2 mm thick samples. These have a larger thickness, providing more material to affect the angle shifts observed between each infill. The 2 mm thick samples have a 97.4% reduction in mass from the prisms. The experimental setup is the same as the 1 mm samples; a polynomial fit was obtained from the 2 mm FEM simulations (Figure 7a). The electric far field was measured for each infill and Δθ was calculated from the maximum reflected angles. Figure 7b compares the measured and simulated angle changes. The index of refraction is determined using the polynomial fit and compared to the prism measurements in Figure 7c.

The measured index of refractions for the 2 mm samples are in better agreement with the prism measurement than for the 1 mm samples. The maximum error is 3% for the 100% infill, and the remaining infills have only an average error of 0.93% compared to the prism measurements.

It is observed that the larger sample thickness improves the accuracy of the index measurements. Two more sample thicknesses were fabricated and measured; this includes the 3 mm and 4 mm designs. However, as the sample thickness increases, Δθ becomes more complex and can no longer be fit to a polynomial due to having multiple index solutions for the majority of the Δθ values, as shown in Figure 8a,b. To eliminate this, a fish-eye lens can be added to the front of the sensor, which will not affect the incoming beam but will exaggerate the reflected beams and change the peak angles produced by the sensor. This approach allowed us to use the same metasurface beam-splitting sensor throughout all of the experiments. The change in Δθ from the presence of the fish-eye lens produced using FEM simulations can be observed in Figure 8c,d.

As can be observed from the FEM results, the addition of a fish-eye lens eliminates the Δθ complexity for the 3 mm and 4 mm sample sizes. This allows for a polynomial fit and the ability to measure the index of refraction using the current beam splitter sensor design for these thicker samples. The presence of the lens does not benefit the 1 mm or 2 mm samples. The fish-eye lens has an inner spherical radius of 45 mm, a total diameter of 140 mm, and a thickness of 48 mm (Figure 9b) and was designed to achieve the maximum scattering angles given the geometry of the beam splitter. The initial lens design was tested using ray tracing, illustrated in Figure 9c. The final lens is 3D printed using natural Polymaker PolyLite PLA at a 50% gyroidal infill to coincide with a measured index of refraction of 1.3. To verify the efficacy of the lens, a comparison was made between the normalized electric field for both the simulated and fabricated lenses (Figure 9d,e). A good agreement can be observed between the measurement and simulation. A modified sensor holder is printed to accommodate the addition of the fish-eye lens and is illustrated in Figure 9a.

The measurements for the last two sample thicknesses, 3 mm, and 4 mm, are performed the same way as the 1 mm and 2 mm measurements, except for adding the 3D printed fish-eye lens. The 3 mm samples have a mass reduction of 96.3% and the 4 mm samples have a reduction of 95.2%. A polynomial fit is obtained from the 3 mm and 4 mm FEM simulations (Figure 10a,d). The electric far field is measured for each infill and Δθ is calculated from the maximum reflected angles. Figure 10b,e compare the measured and simulated angle changes. The index of refraction is determined using the polynomial fit and compared to the prism measurements in Figure 10c,f.

The 3 mm samples have a maximum error of 5% for the 40% infill and an average error of 2% for the remaining samples; the 4 mm samples have a maximum error of 3.8% for the 60% infill and an average error of 1.14% for the remaining samples.

## 6. Conclusions

In our study, we present an alternative approach for determining the refractive index of an unknown sample using a metasurface beam-splitting reflector. The metasurface beam-splitting reflecting sensor was created using PCB milling and 3D printing techniques. The sensor design was established using the Fourier transform method of array synthesis, and its functionality was verified through FEM simulations. The sample’s refractive index is determined by measuring the change in beam deflection angles of the metasurface beam splitter sensor due to the presence of a sample. This arises from the imparted amplitude and phase mismatch due to the sample refractive index at each meta-atom comprising the sensor. These amplitude and phase variations also affect the amplitude of the sensor’s split beams. The extinction coefficient of the sample, that is, the imaginary component of the refractive index, also affects the amplitude of the split beams of the sensor and cannot be discerned from the amplitude reduction due to the refractive index itself.

To ascertain the accuracy of the sensor, we conducted an analysis involving forty-four 3D printed dielectric samples with varying infill percentages and thicknesses. Our results indicated minimal deviations in our measurements across all samples compared to the implementation of Snell’s law. Notably, we observed that samples with a thickness of 2 mm and above yielded the most favorable results. This metasurface reflective sensor requires substantially less sample material when compared to conventional methods, leading to material cost reductions and faster characterization processes. This device could provide a practical alternative to traditional methods for characterizing indices, eliminating the need for large sample sizes and expensive equipment.

## Figures and Tables

**Figure 1 sensors-24-06519-f001:**
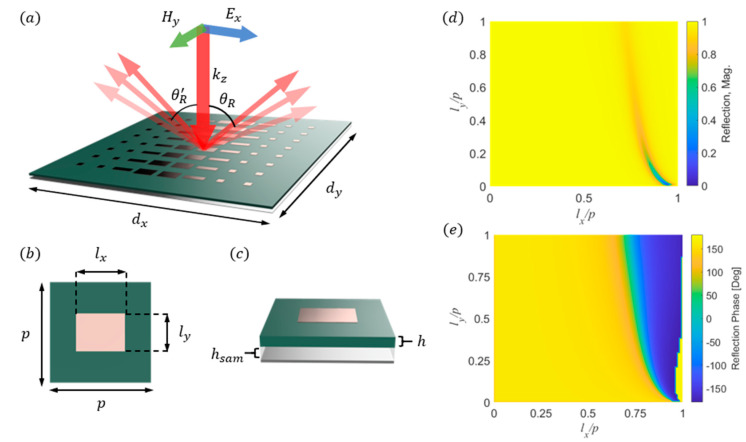
(**a**) Schematic of the metasurface beam splitter reflecting sensor. ($d_{x}$, $d_{y}$) are the sensor’s length and width proportional to p and ($\theta_{R}^{\prime }$, $\theta_{R}$) represent the desired reflected angles. (**b**) Top view of the metasurface unit cell with p=9.8 mm, and $l_{x}$ and $l_{y}$ are the patch dimensions varied from 50 μm to 9.3 mm for FEM simulations. (**c**) Illustrates the side view of the unit cell. h=0.8 mm is the height of the substrate and $h_{sam}=1\;mm$ for the unit cell simulations. The sample height can be varied to accommodate various sample thicknesses. Simulated reflection (**d**) and phase (**e**) surface plots of the metasurface unit cell as a function of normalized lengths ($l_{x}/p$, $l_{y}/p$) of the conductive patches.

**Figure 2 sensors-24-06519-f002:**
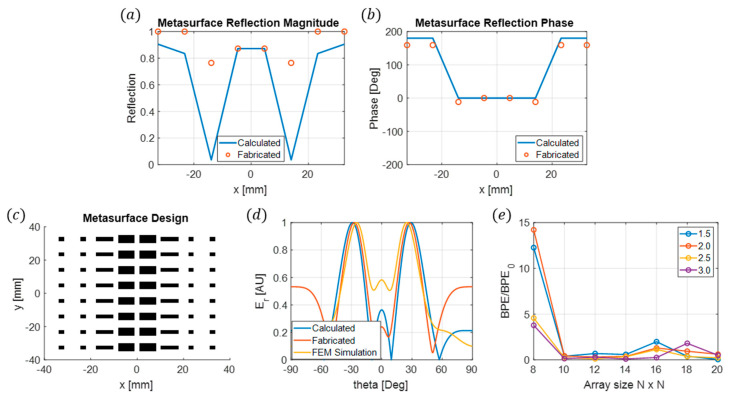
(**a**) Metasurface reflection magnitude and (**b**) reflection phase across the *x*-axis of the beam splitter. The solid line is calculated and the symbols are the nearest-neighbor fit. (**c**) Metasurface patch design from the nearest-neighbor fit. (**d**) A comparison of the electric far field for theory calculations, fabricated design calculations, and FEM simulations for the metasurface beam splitter sensor with a 1 mm air gap between the substrate and reflector. (**e**) Beam pointing error BPE relative to the beam pointing error of a sample-free sensor $BPE_{0}$ as a function of sensor size and sample index.

**Figure 3 sensors-24-06519-f003:**
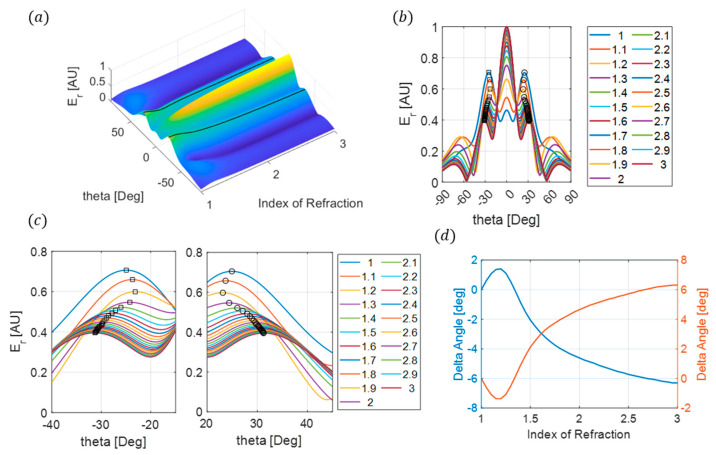
(**a**) Surface plot of the electric-far-field response for varying the sample’s index of refraction and RX angle. The solid line indicates the maximum reflected angles. (**b**) A comparison of the electric far field for varying sample indices. Symbols indicate the location of the maximum reflected angles. (**c**) Close-up of the angle shifts observed for varying sample indices. (**d**) Change in angle from the initial reflected angles, not including a sample. The blue line is for the original −30° and the red line is for the original 30° reflected peaks.

**Figure 4 sensors-24-06519-f004:**
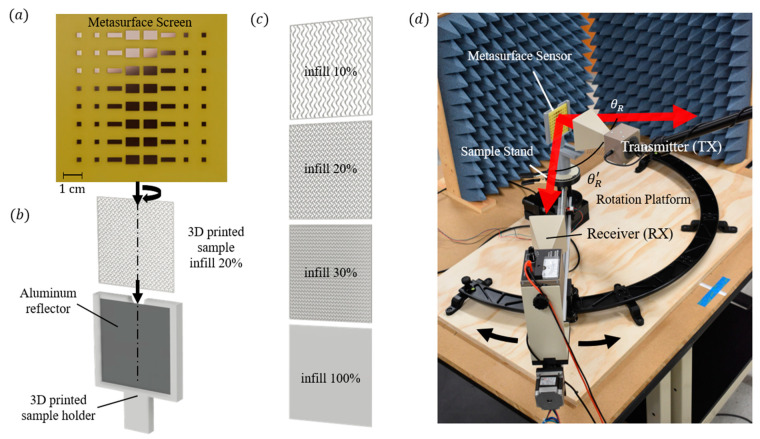
(**a**) Final metasurface beam splitter design etched on a FR4 single-sided copper-clad laminate circuit board. (**b**) Shows the schematic of the sensor holder and how all components fit together, including the aluminum reflector, sample, and metasurface beam splitter. These components slide and are held together using the 3D printed sample holder. The metasurface screen is enlarged to show detail. (**c**) The 3D printed 1 mm samples, from top to bottom, are 10%, 20%, 30%, and 100%. (**d**) Photograph of the experimental setup. This shows the location of the TX and RX horns with ray beams drawn to illustrate hypothetical locations of the reflected beams. The receiver is mechanically moved around the sensor using a motorized rotating optical platform at 0.1° increments ±20° from the measured maximum reflected peaks. It should be noted that the position of the TX horn does not affect the angle measurements due to their locations being outside the region of interference that would be produced by the TX horn when the RX is behind it.

**Figure 5 sensors-24-06519-f005:**
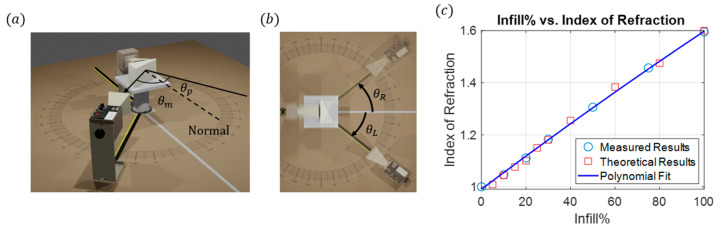
(**a**) Experimental setup for prism index measurements. (**b**) Top view of the experimental setup. The receiving horn (RX) is manually moved through 1° increments, identifying the location of the peak intensity first on the left-hand configuration and then repeated for the right-hand configuration. The average theta value is calculated and used for the final calculations. (**c**) Infill % compared with the index of refraction. The solid blue line is the polynomial fit, circles are the measured index, and squares are the FEM simulated index.

**Figure 6 sensors-24-06519-f006:**
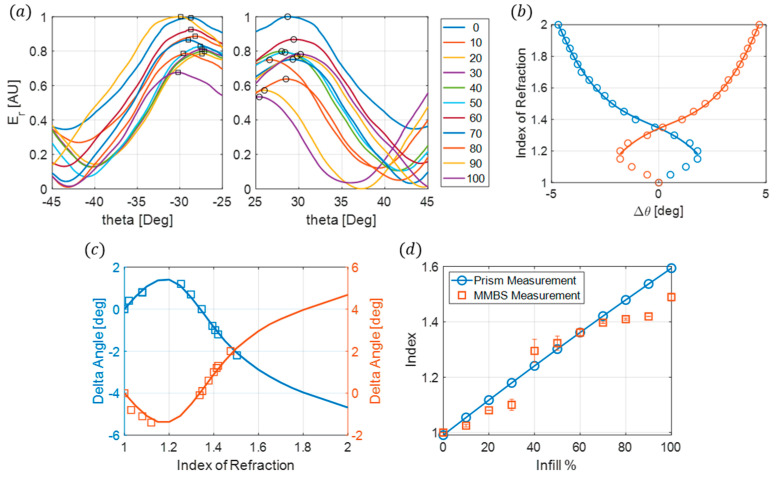
(**a**) Normalized far-field electric field for the 1 mm samples of different infills at both −30° and 30° reflected angles. The symbols represent the location of the maximum angles. (**b**) Index of refraction as a function of Δθ. The symbols are the FEM simulations and the solid line is the polynomial fit. (**c**) Measured delta theta (symbols) compared to FEM simulations (solid lines). The index of refractions of the measured delta theta is determined from the polynomial fit. (**d**) Comparison of the measured index of refraction from the metasurface beam splitter sensor (square symbols) compared with the measured index of refraction from the prism measurements (solid blue line) as a function of infill percentage.

**Figure 7 sensors-24-06519-f007:**
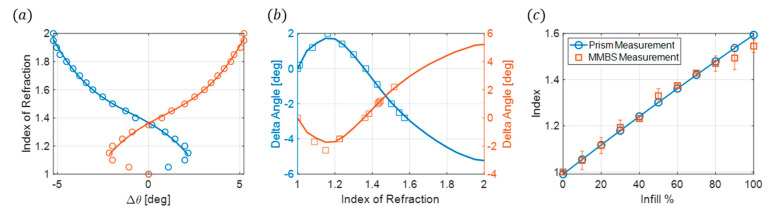
(**a**) Index of refraction as a function of Δθ. The symbols represent FEM simulations and the solid line is the polynomial fit. (**b**) Measured delta theta (symbols) compared to FEM simulations (solid lines). (**c**) Comparison of the measured index of refraction from the metasurface beam splitter sensor (square symbols) compared with the measured index of refraction from the prism measurements (solid blue line) as a function of infill percentage.

**Figure 8 sensors-24-06519-f008:**
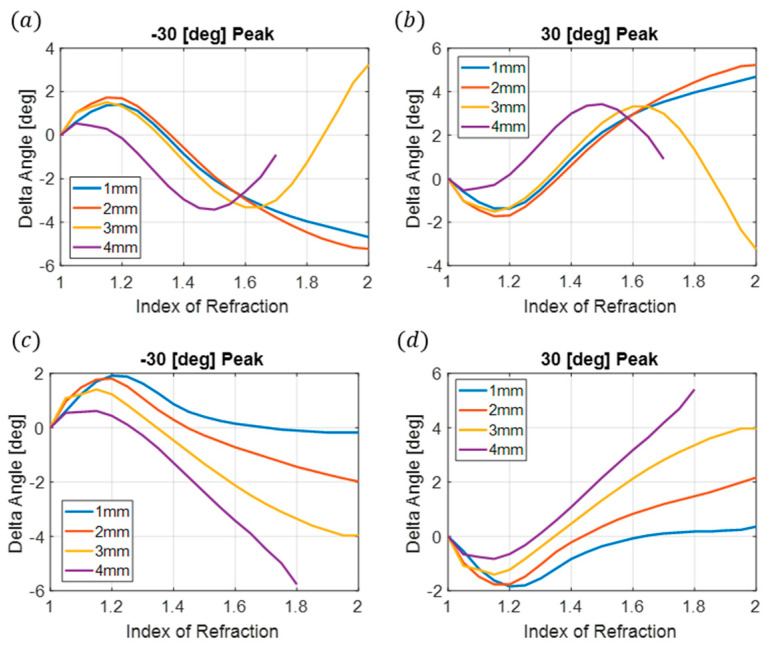
FEM simulations for Δθ as a function of the index for varying sample thickness without the fish-eye lens (**a**,**b**) and including the fish-eye lens (**c**,**d**) for the −30° and 30° peaks, respectively. The 4 mm sample does not converge to two distinct reflected angles after an index of 1.7, but this is slightly improved to 1.8 in the presence of the lens.

**Figure 9 sensors-24-06519-f009:**
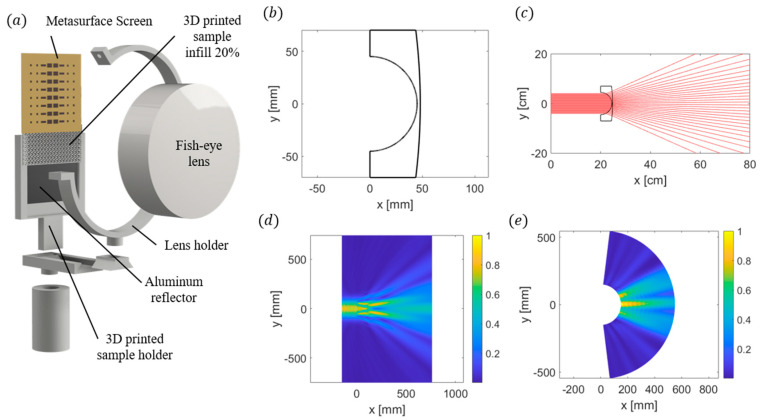
(**a**) Illustration of the modified sensor holder to accommodate the addition of the fish-eye lens. The lens position can be changed using the dovetail track, but it is positioned directly at the beam-splitting surface for the experiments. (**b**) Cross-section of the fish-eye lens design. (**c**) Ray tracing of the initial design to achieve maximum scattering. (**d**) FEM simulation of the normalized electric field for the fish-eye lens. (**e**) Measured normalized electric field for the fabricated fish-eye lens.

**Figure 10 sensors-24-06519-f010:**
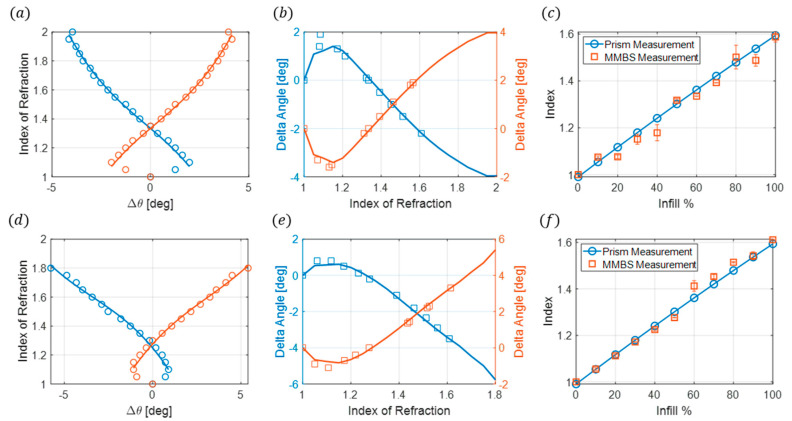
(**a**,**d**) Index of refraction as a function of Δθ. The symbols represent FEM simulations and the solid line is the polynomial fit. (**b**,**e**) Measured delta theta (symbols) compared to FEM simulations (solid lines). (**c**,**f**) Comparison of the measured index of refraction from the metasurface beam splitter sensor (square symbols) compared with the measured index of refraction from the prism measurements (solid blue line) as a function of infill percentage. The top row is the 3 mm samples and the bottom is the 4 mm samples.

## Data Availability

Data underlying the results presented in this paper are not publicly available at this time but may be obtained from the authors upon reasonable request.

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
