# Peer review of "Application of Single-Frequency Arbitrarily Directed Split Beam Metasurface Reflector in Refractive Index Measurements"

_sensors, 2024, doi:10.3390/s24206519_

Round 1

Reviewer 1 Report

Comments and Suggestions for Authors

The paper presents an alternative approach for determining the refractive index 337 of an unknown sample using a metasurface beam-splitting reflector. The design requires substantially less sample material when compared to conventional methods, leading to material cost reductions and faster characterization processes. Overall, the work is of high quality and will benefit researchers who study metasurface and material characterization. Therefore, the paper shall be accepted by the journal. I have the following two points that should be addressed first.

1.  Fig. 4b should be revised, with the metasurface, the sample, and the holder marked by text.

2. How about the tangential loss of the material? As known, almost all the materials are lossy, so how to measure the imaginary part of the complex dielectric parameter?

Reviewer 2 Report

Comments and Suggestions for Authors

Authors present a technique for the determination of a sample refractive index based on a reflection metasurface. In general, the manuscript is well-structured; the design procedure of the metasurface is described well. Presented conclusions supported by experimental and simulation results.

However, there are some questions.

1. Since the proposed method is based on deflected beam detection, one may say that its beamwidth may affect the tolerance of the refractive index value extracted from measurements. Does metasurface area affect tolerance? If so, why did authors use this specific size of metasurface? Is it optimal?

2. As a benefit, authors state that the proposed method allows the use of a small sample compared to other methods. Are there any comparisons in terms of wavelength?

3. Authors demonstrate their prototype operating around 10 GHz. There are a lot of measurement techniques in this range and higher (for example, https://doi.org/10.3390/s24030755, 10.1109/ICCPS.2014.7062271, 10.1109/JSEN.2013.2285918, etc.). Is there any benefit over existing methods? If so, it should be emphasized.

Some minor remarks:

1. Figure 2d - typo in legend. It is not clear what does mean "design" and "numerical", please provide more detailed explanation.

2. In spite of authors explaining in the abstract that samples should be placed between metasurface and metal screen, it would be convenient for readers to mention it in the section describing the holder and/or modify Fig. 4ab to an exploded view.

Round 2

Reviewer 2 Report

Comments and Suggestions for Authors

After revisions the article became more consistent and clear.